# Scapulothoracic Alignment Alterations in Patients with Walch Type B Osteoarthritis: An In Vivo Dynamic Analysis and Prospective Comparative Study

**DOI:** 10.3390/jcm10010066

**Published:** 2020-12-27

**Authors:** Alexandre Lädermann, George S. Athwal, Hugo Bothorel, Philippe Collin, Adrien Mazzolari, Patric Raiss, Caecilia Charbonnier

**Affiliations:** 1Division of Orthopaedics and Trauma Surgery, La Tour Hospital, 1217 Meyrin, Switzerland; mazzolaria@gmail.com; 2Faculty of Medicine, University of Geneva, 1206 Geneva, Switzerland; caecilia.charbonnier@artanim.ch; 3Division of Orthopaedics and Trauma Surgery, Department of Surgery, Geneva University Hospitals, 1205 Geneva, Switzerland; 4Department of Surgery, Roth MacFarlane Hand and Upper Limb Center, St. Joseph’s Health Care London, London, ON N6A 4V2, Canada; gsathwal@hotmail.com; 5Research Department, La Tour Hospital, 1217 Meyrin, Switzerland; hugo.bothorel@latour.ch; 6Centre Hospitalier Privé Saint-Grégoire (Vivalto Santé), 35760 Saint-Grégoire, France; collin@west.bzh; 7Shoulder and Elbow Surgery, OCM (Orthopädische Chirurgie München), Steinerstrasse 6, 81369 Munich, Germany; patric.raiss@gmail.com; 8Medical Research Department, Artanim Foundation, 1217 Meyrin, Switzerland

**Keywords:** shoulder arthritis, B glenoid, scapular malalignment, kinematics, biomechanics, 3D simulation

## Abstract

Background: Kinematic changes of the scapulothoracic joint may influence the relative position of the glenoid fossa and, consequently, the glenohumeral joint. As the alignment of the scapula relative to the thorax differs between individuals, such variability may be another factor in the development of posterior head subluxation. The purpose of this study was to compare scapulothoracic alignment in pathologic type B shoulders with contralateral healthy shoulders. Methods: Seven adult volunteers with unilateral type B glenohumeral osteoarthritis (OA) underwent bilateral computed tomography (CT) scans of the shoulders and arms. A patient-specific, three-dimensional measurement technique that coupled medical imaging (i.e., CT) and optical motion capture was used. Results: The scapulothoracic distance at the trigonum was 75 ± 15 mm for pathologic shoulders and 78 ± 11 mm for healthy shoulders (*p* = 0.583), while at the inferior angle, it was 102 ± 18 mm for pathologic shoulders and 108 ± 12 mm for healthy shoulders (*p* = 0.466). Conclusion: Scapula positioning at a resting position did not differ between pathologic and healthy shoulders. However, pathologic shoulders tended to be limited in maximal glenohumeral motion and exhibited greater anterior tilt of the scapula in internal rotation at 90 degrees, which may be adaptive to the restricted glenohumeral motion.

## 1. Introduction

Type B glenohumeral osteoarthritis (OA) is believed to be initiated by progressive posterior humeral head subluxation. The exact cause of this posterior translation is as yet unknown and is likely to be multifactorial. The identified associations include altered glenoid version [1], decreased humeral torsion [2], and variations in proximal humeral and acromial morphologies [3]. Aleem et al. [4] suggested that posterior subluxation could be the result of bone adaptation induced by periscapular muscle imbalance. Other hypotheses have been stated but not confirmed, including the morphology of the humeral head [5] or repetitive dynamic posterior subluxations [6].

The scapula is linked to the axial skeleton via the acromioclavicular joint and 17 muscular attachments [7]. Kinematic changes of the scapulothoracic joint may influence the relative position of the glenoid fossa and, consequently, the glenohumeral joint. As the alignment of the scapula relative to the thorax differs between individuals, such variability may be another factor in the development of posterior humeral head subluxation and subsequent erosion patterns. The purpose of this study was to evaluate and compare the scapulothoracic alignment in pathologic shoulders with contralateral healthy shoulders. We hypothesized that altered scapulothoracic alignments would be observed when comparing type Bs with the contralateral normal side. The results of this study may improve our understanding of glenohumeral pathoanatomy and posterior glenoid erosion patterns, as well as potentially assist with OA prevention.

## 2. Experimental Section

### 2.1. Subjects

The authors prospectively enrolled seven adult volunteers willing to undergo testing between July and November 2019. The subjects were being treated by the primary author (A.L.) for unilateral glenohumeral OA with a Walch type B glenoid (Figure 1). The exclusion criteria were: (i) previous shoulder surgery, (ii) spinal column deformity, (iii) psychiatric problems that precluded informed consent or inability to read or write, (iv) bilateral symptoms or disease, and (v) incomplete documentation.

### 2.2. Ethical Approval

The study protocol was approved by the hospital ethics committee (AMG, Association des Médecins du Canton de Genève, Ethic Commission for Clinical Research: Protocole 12–18), and all patients gave written informed consent before participating in the study.

### 2.3. Outcomes

The outcomes of interest were the initial alignment of the scapula relative to the thorax, defined by the scapulothoracic distance at resting position (arm at side in adduction and neutral rotation), as well as maximal glenohumeral and scapulothoracic motions for different movements (specific motions outlined in Section 2.6 below).

### 2.4. Radiographic Evaluation

All volunteers underwent standardized computed tomography (CT) of bilateral shoulders and arms in the supine position with arms placed along the body. The CT examinations were conducted with a LightSpeed VCT 64 rows system (General Electric Healthcare, Milwaukee, WI, USA). Images were acquired at a 0.63 mm slice thickness. The CT images were used to create patient-specific, three-dimensional (3D) models of the shoulder bones (humerus, scapula, clavicle, and sternum) in the Mimics software program (Materialize NV, Leuven, Belgium).

### 2.5. Scapula Positioning Relative to the Thorax

The position of the scapula relative to the thorax was calculated using the CT images for both sides based on two measurements; firstly, the shortest distance in millimeters between the most medial point of the trigonum spinae of the scapula and the adjacent spinous process, and secondly, the shortest distance between the most medial point of the inferior angle of the scapula and the adjacent spinous process (Figure 2).

### 2.6. Motion Capture

All patients participated in a motion capture session, where both pathologic and contralateral normal shoulders were analyzed. Kinematic data were recorded using a Vicon MX T-Series motion capture system (Vicon, Oxford Metrics, U.K.) consisting of 24 T40S cameras sampling at 120 Hz. The patients were equipped with a previously described shoulder marker protocol [8], which included 69 spherical retroreflective markers. The setup included four markers (Ø 14 mm) on the thorax (sternal notch, xyphoid process, and C7 and T8 vertebra), four markers (Ø 6.5 mm) on the clavicle, four markers (Ø 14 mm) on the upper arm—two placed on the lateral and medial epicondyles and two as far as possible from the deltoid—and 57 markers on the scapula (1 × Ø 14 mm on the acromion and a 7 × 8 grid of Ø 6.5 mm). Finally, additional markers were placed over the body (on the other arm and the legs) to provide a global visualization of motion. The same biomechanical motion capture specialist (C.C.) attached all markers and performed all measurements.

For each shoulder under investigation, the patients were asked to perform the following motor tasks three times (Figure 3): (1) internal–external rotation with an approximately 90 degree abduction and the elbow flexed 90 degrees (IR90 degrees, ER90 degrees); (2) empty can abduction from neutral to maximum abduction in the scapular plane; (3) three daily activities (crossing arms, hand behind back, and combing hair). The maximum range of motion (ROM) of three trials for each movement was evaluated by the same investigator (C.C.). Only the median value among the three evaluated ROMs was used for statistical analyses to avoid potential effects of outliers.

### 2.7. Kinematic Analysis

Shoulder kinematics were computed from the recorded marker trajectories using a validated biomechanical model, which accounted for skin motion artefact [8,9]. The model was based on a patient-specific kinematic chain using the shoulder 3D models reconstructed from the CT data and a global optimization algorithm with loose constraints on joint translations (accuracy: translational error < 3 mm and rotational error < 4 degrees). Figure 3 and Figure 4 show, respectively, examples of computed postures and the positioning of the markers around the shoulder.

To permit motion description of the shoulder kinematic chain, local coordinate systems were established based on the definitions suggested by the International Society of Biomechanics (ISB) [10] to represent the thorax, clavicle, scapula, and humerus segments. They were created using anatomical landmarks identified on the patient’s bony 3D models. The glenohumeral joint center was calculated based on a sphere fitting method [11].

### 2.8. Range of Motion

The maximal glenohumeral ROM was quantified for abduction, IR90 degrees, and ER90 degrees. This was obtained by calculating the relative orientation between the scapula and humerus coordinate systems and then expressed in clinically recognizable terms (flexion/extension, abduction/adduction, and IR/ER) [12].

The maximal scapulothoracic ROM was evaluated for the same movements (abduction, IR90 degrees, and ER90 degrees), as well as the three daily activities (crossing arms, hand behind back, and combing hair). This was achieved with the same method using the thorax and scapula coordinate systems. The relative orientation of the scapula with respect to the thorax was decomposed in three successive rotations according to the ISB standards: protraction/retraction, lateral/medial rotation, and posterior/anterior tilt (Figure 5).

### 2.9. Statistical Analysis.

Assuming that a difference of 20 ± 10 mm in scapulothoracic distance between healthy and B glenoid shoulders is clinically relevant [13], a minimum of five patients were required for this study (statistical power of 0.80, and level of significance (alpha) at 0.05).

The Shapiro–Wilk test was used to check the normality of the distributions. Descriptive statistics are presented as means and standard deviations (SDs). For normally distributed quantitative data, the significance of the kinematic differences among groups (healthy versus pathologic shoulders) was determined using the paired Student *t*-test. Conversely, for quantitative data that were not distributed normally, the significance among groups was determined using the Wilcoxon signed-rank test. Statistical analyses were performed using R version 3.6.2 (R Foundation for Statistical Computing, Vienna, Austria). *p*-values < 0.05 were considered statistically significant.

## 3. Results

The study group was a mean of 44 ± 12 years old (median 45 years; range 19–57 years), and consisted of two women (29%) and five men (71%). The mean weight was 76 ± 13 kg (median 78 kg; range 58–90 kg) and the mean height was 176 ± 8 cm (median 180 cm; range 165–183 cm). The involved shoulder (Walch type B) was mostly on the dominant side (71% vs. 29%, *p* = 0.286). Two patients had unexploitable corrupted measurements of glenohumeral and scapulothoracic motions for the contralateral shoulder, leaving five patients for analyses in the abduction, crossing arms, hand behind back, and combing hair movements, as well as five patients in IR90 degrees and ER90 degrees.

### 3.1. Scapula Positioning Relative to the Thorax

The resting positions of the scapula for Walch type B osteoarthritic shoulders were not significantly different to contralateral normal shoulders (*p* < 0.583). The scapulothoracic distance at the trigonum, comparing type B to normal, was 75 ± 15 mm and 78 ± 11 mm (*p* = 0.583), respectively, and at the scapula inferior angle was 102 ± 18 mm and 108 ± 12 mm (*p* = 0.466), respectively (Table 1).

### 3.2. Glenohumeral Motion

The maximal glenohumeral ROM tended to be lower for pathologic shoulders compared with healthy shoulders, in both abduction (73 ± 21 degrees vs. 94 ± 11 degrees, *p* = 0.074) and IR90 degrees (23 ± 13 degrees vs. 32 ± 10 degrees, *p* = 0.128), although not reaching statistical significance (Table 2).

### 3.3. Scapulothoracic Motion

Both pathologic and healthy shoulders displayed lateral rotation for all movements initiated from the resting position (absence of medial rotation). The maximal scapulothoracic ROM was significantly different between pathologic and healthy sides, with a greater anterior tilt of the scapula in IR90 degrees in pathologic shoulders (26 ± 8 degrees vs. 10 ± 5 degrees, *p* = 0.025). The same observational trend was made for crossing arm movements, although this difference was not statistically significant (40 ± 9 degrees vs. 11 ± 24 degrees, *p* = 0.073).

## 4. Discussion

The principal finding of this study was that alignment of the scapula relative to the thorax at the resting position was not substantially different between Walch type B and healthy shoulders, with comparable scapulothoracic distances at the scapula trigonum spinae and inferior angle regions. However, the kinematic analyses revealed that pathologic shoulders displayed a greater anterior tilt (by 15 degrees) in IR90 degrees compared to healthy shoulders.

The etiology of progressive posterior glenoid erosion in the Walch type B is as yet unknown and is most likely to be multifactorial. Several authors have reported that posterior glenoid erosion could be due to specific anatomic features (excessive glenoid retroversion, humeral retrotorsion, acromial roof, or proximal humeral morphologies) [1,2,3,5] or other factors such as repetitive dynamic posterior subluxations [6] or periscapular muscle imbalance [4]. In the present study, shoulders with posterior glenoid erosions tended to be limited in IR90 degrees and abduction, while displaying greater anterior tilt of the scapula in IR90 degrees and crossing arm movements. These findings suggest a possible association between scapula alignment during certain shoulder movements and posterior glenoid erosion. In our patient cohort, we theorize that restrictions in glenohumeral motion, specifically internal rotation, can be compensated to a certain amount by increased scapulothoracic mobility such as anterior tilt.

Borich et al. [14] reported, in a cohort of overhead athletes, a low but significant correlation between scapula anterior tilt and a deficiency in glenohumeral internal rotation. Participants who demonstrated a deficiency of at least 20% of internal rotation had a greater anterior tilt of the scapula in IR90 degrees compared with the control group (25 ± 12 degrees vs. 15 ± 12 degrees). Interestingly, these results are comparable to our findings, with 2 ± 6 degrees for pathologic shoulders compared with 10 ± 5 degrees for healthy shoulders in IR90 degrees.

As there is no direct bony articulation between the scapula and the thorax, we hypothesized that periscapular muscle imbalance would trigger kinematic changes of the scapula, which would thereafter influence the position of the glenoid cavity and affect the glenohumeral joint. While it was impossible to affirm the cause and effect relationship between scapula dyskinesis and glenoid OA, the authors believe that the malalignment of the scapula relative to the thorax during certain movements was an indirect consequence of glenoid erosion, rather than one of its direct origins. Further studies are therefore needed to investigate such biomechanical relationships by following the impact of glenoid OA evolution on scapula kinematic changes, and vice versa. Additionally, the greater anterior tilt, to initially compensate for diminished glenohumeral motion, could eventually be a potentiator posteroinferior erosion due to malalignment.

An excessive anterior tilt of the scapula considerably reduces the subacromial space, and could thereby lead to different pathologies in the long term, including subacromial bursitis, rotator cuff tendinopathy, or full-thickness rotator cuff tears [15,16,17]. Furthermore, it has been reported that a greater anterior tilt of the scapula is associated with a considerable posterior capsule tightening, which could also trigger pain [18]. Interestingly, most of our patients reported clinical symptoms (pain or stiffness) during shoulder movements, which could be related to the aforementioned pathologies. It was difficult to know if improving or correcting the abnormal scapular tilt would lead to improved patient outcomes, or potentiate limitations. As such, future directions of study would need to assess the role of physiotherapy in correcting the pathologic scapular position and its impact on patient outcomes.

We acknowledge the following limitations in our study. First, the accuracy of the kinematics computation from motion capture data. Glenohumeral orientation errors were within 4 degrees, which is acceptable for clinical use in the evaluation of shoulder pathology. Second, the arm dominance may represent a potential confounding factor, as some studies have reported scapular kinematic differences between dominant and non-dominant shoulders [19,20]. However, such asymmetry is often reported in unilateral sport athletes, who were not evaluated in our study. Finally, the sample size evaluated in this study was relatively small. This study might therefore be underpowered to analyze ROM differences between healthy and pathologic shoulders. However, we believe that these limitations do not call into question the results of this study. To the best of our knowledge, this non-invasive research was the first to calculate both glenohumeral and scapulothoracic motions based on skin markers. Future studies with larger sample sizes should analyze various shoulder pathologies to better understand the precise role of the scapulothoracic joint that is probably underestimated.

## 5. Conclusions

It was theorized that scapulothoracic malalignment may be present in Walch type B osteoarthritis shoulders. The results of this study found no statistically significant differences in the resting scapulothoracic position between osteoarthritic and contralateral normal shoulders. However, Walch type B shoulders had some limitations in maximal glenohumeral motion, but exhibited significantly greater anterior scapular tilt with internal rotation, which may be adaptive.

## Figures and Tables

**Figure 1 jcm-10-00066-f001:**
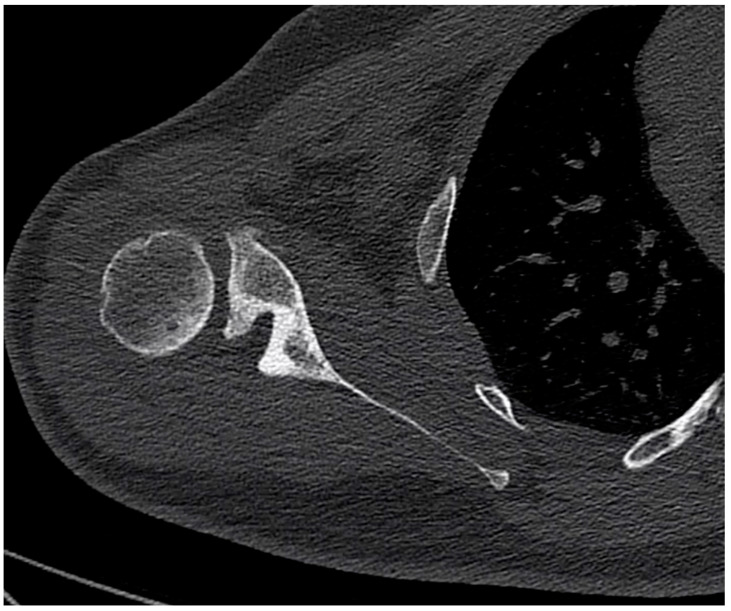
Example of a right B2 glenoid. Observe the biconcave glenoid and the posterior subluxation of the humeral head.

**Figure 2 jcm-10-00066-f002:**
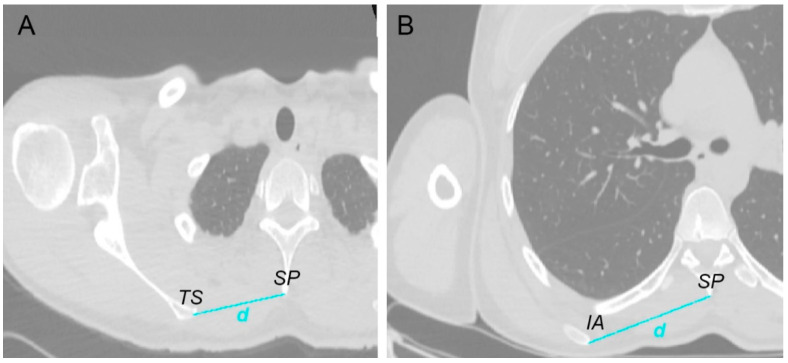
Measurement of the scapula positioning relative to the thorax using computed tomography (CT) images: (**A**) shortest distance *d* between the most medial point of the trigonum spinae (*TS*) of the scapula and the adjacent spinous process (*SP*); (**B**) shortest distance *d* between the most medial point of the inferior angle (*IA*) of the scapula and the adjacent spinous process (*SP*).

**Figure 3 jcm-10-00066-f003:**
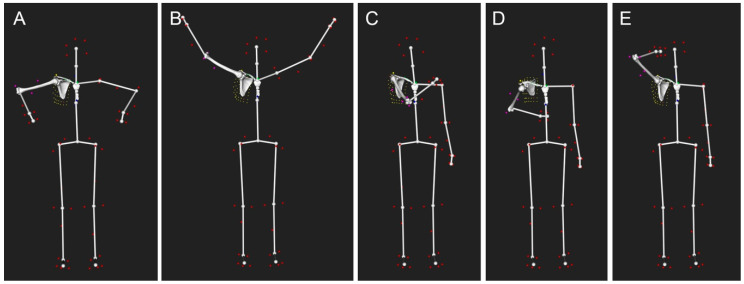
Examples of computed postures from motion capture (right shoulder): (**A**) maximum internal rotation with a 90 degree abduction and the elbow flexed 90 degrees (IR90 degrees); (**B**) maximum abduction in the scapular plane; (**C**) crossing arms; (**D**) hand behind back; (**E**) combing hair. The colored dots represent the retro-reflective skin markers. A virtual skeleton was also used to better visualize the global motion.

**Figure 4 jcm-10-00066-f004:**
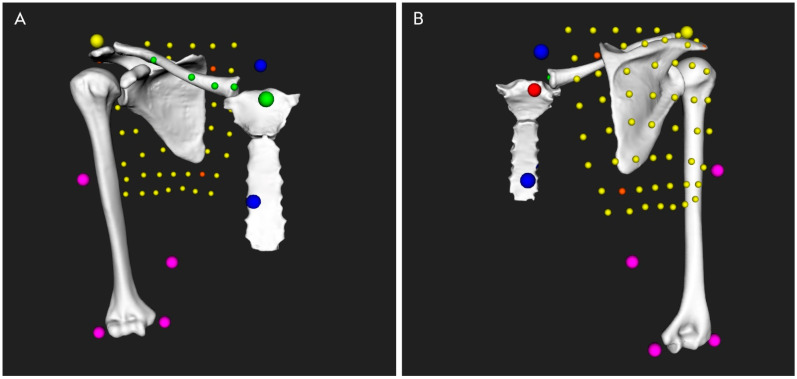
Front (**A**) and back (**B**) views of the markers setup on the shoulder (right shoulder). The colored dots represent the retro-reflective skin markers.

**Figure 5 jcm-10-00066-f005:**
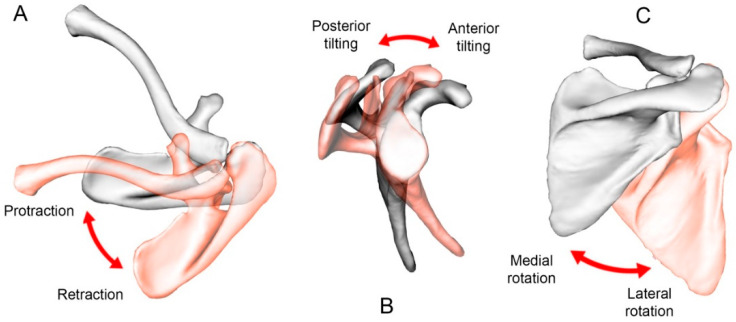
Motion of the scapula relative to the thorax: (**A**) protraction/retraction, (**B**) posterior/anterior tilt, and (**C**) lateral/medial rotation.

**Table 1 jcm-10-00066-t001:** Scapulothoracic distances.

	Healthy Side (*n* = 7 Shoulders)			Pathologic Side (*n* = 7 Shoulders)			
	Mean ± SD	Median	Range	Mean ± SD	Median	Range	*p*-Value
Scapulothoracic distance (mm)							
At the trigonum spinae region	77.7 ± 10.7	74.2	(62.9–95.7)	74.6 ± 14.9	67.4	(62.5–103.5)	0.583
At the inferior scapula angle region	107.9 ± 12.3	108.7	(82.1–119.6)	102.1 ± 18.5	103.1	(78.0–134.1)	0.466

**Table 2 jcm-10-00066-t002:** Glenohumeral and scapulothoracic motions.

	Healthy Side (*n* = 6 Shoulders)				Pathologic Side (*n* = 6 Shoulders)				
	Mean	±SD	Median	Range	Mean	±SD	Median	Range	*p*-Value *
**Abduction (degrees)**									
Glenohumeral motion	94	±11	95	(80–106)	73	±21	67	(45–101)	0.074
Scapular protraction (+)/retraction (–)	−8	±8	−10	(−17 to −7)	−6	±9	−3	(−24 to 2)	0.763
Scapular medial (+)/lateral (–) rotation	−44	±5	−45	(−50 to −37)	−41	±10	−45	(−48 to −22)	0.465
Scapular posterior (+)/anterior (–) tilt	−1	±15	4	(−30 to −11)	−5	±22	4	(−38 to 15)	0.705
**IR90 degrees ****									
Glenohumeral motion (degrees)	32	±10	27	(22–45)	23	±13	20	(8–42)	0.128
Scapular protraction (+)/retraction (–)	9	±13	11	(−11 to 23)	−7	±19	−11	(−24 to 25)	0.321
Scapular medial (+)/lateral (–) rotation	−33	±5	−32	(−41 to −29)	−24	±6	−27	(−29 to −15)	0.086
Scapular posterior (+)/anterior (–) tilt	−10	±5	−8	(−18 to −7)	−26	±8	−23	(−37 to −18)	0.025
**ER90 degrees ****									
Glenohumeral motion	38	±14	36	(25–60)	39	±16	47	(15–53)	0.889
Scapular protraction (+)/retraction (–)	−5	±3	−5	(−8 to −1)	−2	±13	−1	(−20 to 17)	0.661
Scapular medial (+)/lateral (–) rotation	−37	±6	−37	(−46 to −29)	−34	±7	−34	(−40 to −25)	0.499
Scapular posterior (+)/anterior (–) tilt	−4	±9	0	(−19 to 1)	−15	±14	−15	(−32 to 7)	0.200
**Cross arms (degrees)**									
Scapular protraction (+)/retraction (–)	7	±10	5	(−4 to 26)	−4	±9	−2	(−17 to 8)	0.209
Scapular medial (+)/lateral (–) rotation	−33	±8	−32	(−46 to −24)	−31	±4	−31	(−37 to −26)	0.443
Scapular posterior (+)/anterior (–) tilt	−11	±24	−15	(−47 to 20)	−40	±9	−43	(−47 to −21)	0.073
**Hand behind back (degrees)**									
Scapular protraction (+)/retraction (–)	28	±25	37	(−22 to 45)	−18	±26	−28	(−34 to 35)	0.082
Scapular medial (+)/lateral (–) rotation	−17	±7	−17	(−26 to −9)	−13	±6	−13	(−19 to −5)	0.241
Scapular posterior (+)/anterior (–) tilt	−21	±6	−20	(−31 to −14)	−25	±7	−22	(−38 to −18)	0.319
**Comb hair (degrees)**									
Scapular protraction (+)/retraction (–)	−8	±12	−8	(−22 to 9)	−3	±3	−3	(−7 to 2)	0.361
Scapular medial (+)/lateral (–) rotation	−42	±5	−43	(−49 to −35)	−41	±5	−43	(−46 to −34)	0.755
Scapular posterior (+)/anterior (–) tilt	0	±16	2	(−30 to 17)	6	±17	14	(−26 to 17)	0.464

* *p*-values in bold indicate significant differences (*p* < 0.05); ** analyses based on measurements of 5 patients.

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
