# Peer review of "Scapulothoracic Alignment Alterations in Patients with Walch Type B Osteoarthritis: An In Vivo Dynamic Analysis and Prospective Comparative Study"

_jcm, 2020, doi:10.3390/jcm10010066_

Round 1

Reviewer 1 Report

The study is generally well planned and also well done. congratulation!

Such a great topic requires always further interest.

Although the sample size evaluated in this study was relatively small, the methodology to calculate both glenohumeral and scapulothoracic motions based on skin markers is adequately explained and the access is promising.

Line 43: Please correct spelling „ Rogation „

68: Subjects: I think the spinal alignement is very important to be considered here. what about including a spinal coloumn deformity here as an exclusion criteria?

Line 83: Radiographic Evaluation:

CT is still an investigation in the supine position which can be easily influenced by the arm position, patient's weight and the underlying surface which can influence the position of the scapula. that should be standarized in all patients.

Line 145: What about including scapula movements in elevation and depression?

Author Response

Line 43: Please correct spelling „ Rogation „

Done

68: Subjects: I think the spinal alignement is very important to be considered here. what about including a

Line 43: Please correct spelling „ Rogation „

68: Subjects: I think the spinal alignement is very important to be considered here. what about including a spinal coloumn deformity here as an exclusion criteria?

Line 83: Radiographic Evaluation:

Reviewer 1 :

Line 43: Please correct spelling „ Rogation „

Done

68: Subjects: I think the spinal alignement is very important to be considered here. what about including a spinal coloumn deformity here as an exclusion criteria?

We agree with this comment. This information has been added.

Line 83: Radiographic Evaluation:

CT is still an investigation in the supine position which can be easily influenced by the arm position, patient's weight and the underlying surface which can influence the position of the scapula that should be standarized in all patients.

We agree with this comment. The CT were standardized in all patients. This information has been added.

Line 145: What about including scapula movements in elevation and depression?

We took into account the scapula coordinate system according to Wu (Journal of Biomecanics 2005) that only defined pro/retraction, lateral/medial rotation and AP tilt.

Reviewer 2 Report

Clear and well written with very minor typos. Decent methodology.

I would recommend reporting the anthropometric parameters (weight, height) of the patients, if they are available. It could be useful for correlating data found in this article with future studies.

A figure showing Walch's glena type B may be helpful

In figure 2 the colored dots are poorly seen. Another figure might be useful to better illustrate skin markers and better explain kinematic analysis.

The section on future perspectives could be expanded. How the results obtained can guide future studies? A larger sample of patients could allow for more consistent results?
